# Benchmarking Multimodal Mathematical Reasoning with Explicit Visual Dependency

## Abstract

Recent advancements in Large Vision-Language Models (LVLMs) have significantly enhanced their ability to integrate visual and linguistic information, achieving near-human proficiency in tasks like object recognition, captioning, and visual question answering. However, current benchmarks typically focus on knowledge-centric evaluations that assess domain-specific expertise, often neglecting the core ability to reason about fundamental mathematical elements and visual concepts. We identify a gap in evaluating elementary-level math problems, which rely on explicit visual dependencies-requiring models to discern, integrate, and reason across multiple images while incorporating commonsense knowledge, all of which are crucial for broader AGI capabilities. To address this gap, we introduce VCBench, a comprehensive benchmark for multimodal mathematical reasoning with explicit visual dependencies. VCBench includes 1,720 problems across six cognitive domains, featuring 6,697 images (averaging 3.9 per question) to ensure multi-image reasoning. We evaluate 26 state-of-the-art LVLMs on VCBench, revealing substantial performance disparities, with even the top models unable to exceed 50% accuracy. Our findings highlight the ongoing challenges in visual-mathematical integration and suggest avenues for future LVLM advancements.

## 1 Introduction

Recent advancements in Large Vision-Language Models (LVLMs) Anthropic (2025); Deepmind (2025); OpenAI et al. (2024); Bai et al. (2023) have made significant strides in bridging the gap between visual understanding and language processing. These models have achieved remarkable performance across a range of tasks, demonstrating near-expert human-level proficiency in domains such as object recognition, caption generation, and visual question answering Lin et al. (2015); Agrawal et al. (2016). Among the various domains explored, LVLMs have shown particular promise in tasks that require both visual and linguistic reasoning, making them increasingly relevant for real-world applications.

While many visual mathematics benchmarks, such as MathVista Lu et al. (2023) and MathVision Wang et al. (2024a), focus on knowledge-centric evaluations that assess domain-specific mathematical or geometric expertise, they often fail to evaluate a model's core ability to perceive and reason about fundamental mathematical elements and visual concepts. Moreover, these knowledge-centric evaluations are easily influenced by the pre-existing knowledge embedded in large language models, which may obscure true reasoning capabilities. To advance towards Artificial General Intelligence (AGI), a more holistic approach to multi-modal reasoning is needed-one that goes beyond task-specific benchmarks and better captures generalizable cognitive abilities.

In this context, we identify a gap in the evaluation of models on elementary-level math problems Cobbe et al. (2021a); Wei et al. (2023b). These problems, typically at the elementary school level, do not require complex mathematical or geometric reasoning but rely heavily on explicit visual dependencies-the ability to discern and integrate visual features across images and understand how different visual elements relate to one another to solve problems. This mirrors the cognitive development of children, who rely on similar skills to solve problems despite not yet possessing advanced reasoning abilities. Understanding and modeling this form of reasoning is crucial, as it represents a fundamental cognitive ability essential for advancing toward broader AGI capabilities.

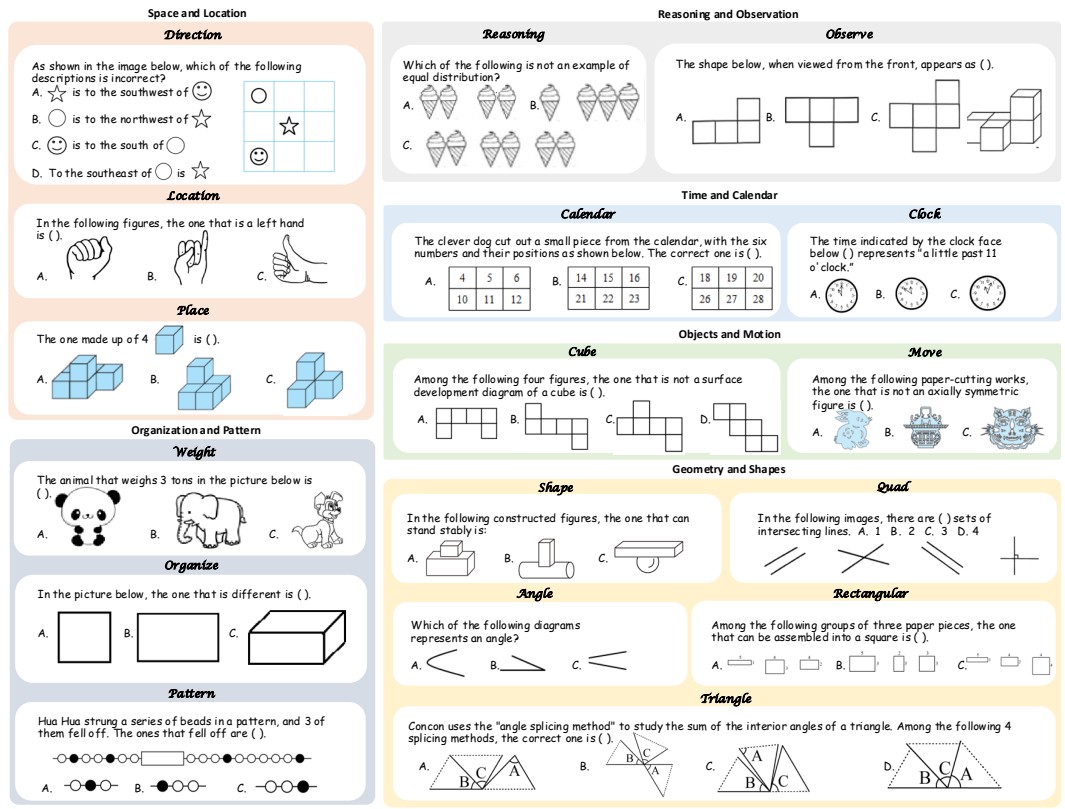

Figure 1: Representative examples from the VCBENCH, showcasing diverse question types and categories including Space and Location (Direction, Location and Place), Reasoning and Observation (Reasoning and Observe), Time and Calendar (Calendar and Clock), Objects and Motion (Cube and Move), Organization and Pattern (Weight, Organize and Pattern), and Geometry and Shapes (Shape, Quad, Angle, Rectangular and Triangle).

To address this gap, we introduce VCBENCH, a comprehensive benchmark designed to assess multimodal mathematical reasoning tasks with explicit visual dependencies. Specifically targeting elementary-level math problems (grades 1–6), VCBENCH focuses on tasks that require reasoning across multiple images to derive solutions. As shown in Figure 1, it covers six key cognitive domains: Time and Calendar, Spatial and Positional Awareness, Geometry and Shapes, Objects and Motion, Reasoning and Observation, and Organization and Patterns. It also evaluates five competencies: temporal reason-

Table 1: Comprehensive Statistics of the VCBENCH Dataset, Including Detailed Breakdown of Question-Image Pairs, Image Distribution, and Question Length Metrics.

| | |
|---|---|
| Examples (Q&A pairs) | 1,720 |
| Images | 6,697 |
| Avg. images per question | 3.9 |
| Avg. question length | 136.2 |
| Max. # images in question | 18 |
| Min. # images in question | 2 |

ing, geometric reasoning, logical reasoning, spatial reasoning, and pattern recognition. These competencies span a broad spectrum, from basic temporal and spatial understanding to more advanced geometric and logical reasoning, providing a thorough evaluation of multimodal model performance. Comprising 1,720 QA pairs and 6,697 images (averaging 3.9 images per question), VCBENCH ensures models must reason across multiple visual inputs, rather than relying on single-image comprehension. With this holistic framework, VCBENCH serves as a valuable resource for advancing research in multimodal mathematical reasoning.

In our extensive experimental evaluation, we assessed 26 state-of-the-art LVLMs across 17 distinct task categories within VCBENCH. Despite achieving near-perfect accuracy on normal human-level performance, the best-performing visual models were unable to exceed 50% accuracy. Many of these

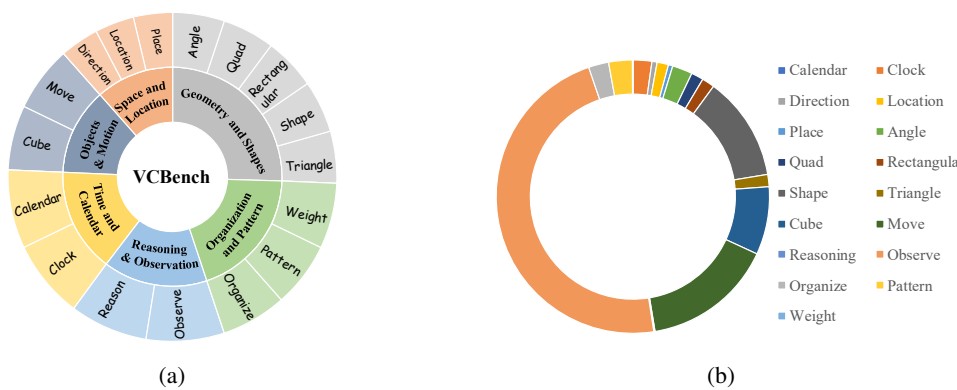

(a)                          (b)

Figure 2: (a) Overview of the VCBENCH dataset structure, highlighting its six main categories and associated subcategories, designed to assess multimodal reasoning capabilities of LVLMs. (b) Distribution of question types in the VCBENCH, illustrating the relative frequency across different visual reasoning subcategories

state-of-the-art models exhibited a notable lack of pattern recognition in images, especially when it came to reasoning tasks that required integrating visual cues across multiple images. Interestingly, we observed that these same tasks could be easily answered by normal human. This highlights a significant gap in current benchmarks, which fail to adequately assess vision-centric mathematical reasoning abilities.

We make several key contributions with VCBENCH:

Unlike existing benchmarks that focus on knowledge-centric evaluations, we emphasize vision-centric assessments. VCBENCH targets problems that do not require specialized knowledge but rely on the common perceptual reasoning of mathematical images and concepts. This approach aligns with the way children learn-first mastering visual reasoning and later acquiring domain-specific knowledge.

VCBENCH is designed around multi-image tasks, with each question containing an average of 3.9 images. This requirement challenges models to explicitly integrate visual cues across multiple images and reason about how they interact, which better reflects real-world scenarios where information is often distributed across multiple visual inputs.

Our benchmark provides a holistic evaluation of various visual reasoning capabilities, such as temporal reasoning, spatial understanding, and pattern recognition. While these tasks may seem simple to children, they represent fundamental reasoning abilities that LVLMs often struggle with. Our experiments demonstrate that tasks considered easy for children-such as identifying time sequences or spatial relationships-prove challenging for state-of-the-art LVLMs, highlighting the gaps in current multimodal reasoning capabilities.

## 2 RELATED WORK

**Large Vision-Language Models.** Large Vision-Language Models (LVLMs) have significantly advanced the integration of vision and language, demonstrating strong performance in tasks such as image captioning, visual question answering (VQA), and complex multimodal reasoning Wang et al. (2024b); Wu et al. (2023). Recent developments, such as Gemini-2.0 Deepmind (2025), QVQ Team (2024), and Calude-3.7-Sonnet Anthropic (2025), showcase emergent abilities in cross-modal instruction-following and chain-of-thought reasoning.

Despite these advancements, quantitatively evaluating LVLMs, particularly in visual mathematical reasoning, remains challenging. Existing benchmarks like MathVista Lu et al. (2023), MathBench Liu et al. (2024), and Math-LLMs Liu et al. (2023) typically assess models within narrow domains, such as arithmetic word problems or geometry-based visual environments. Consequently, these benchmarks primarily measure foundational skills like geometric or spatial reasoning, limiting their

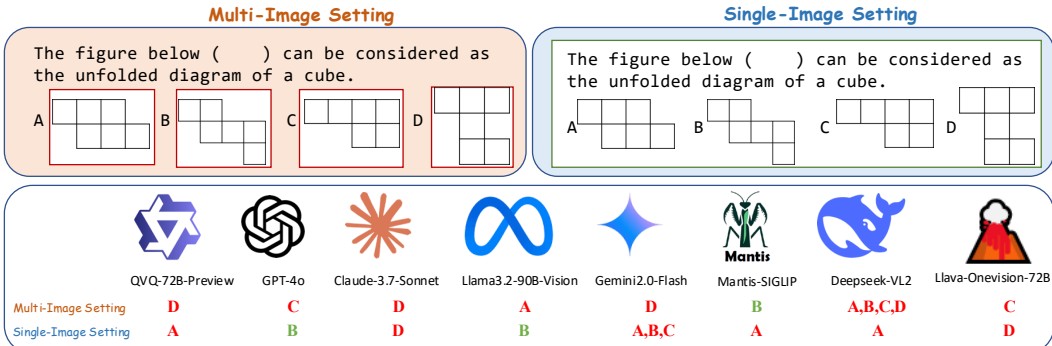

Figure 3: Comparative evaluation of various LVLMs under Multi-Image and Single-Image settings for the same question. The letters (A, B, C, D) indicate models' predictions, with correct answers marked in green and incorrect answers in red.

capacity to comprehensively evaluate broader cognitive integration and reasoning abilities. To address this limitation, we introduce VCBENCH, a systematic evaluation framework designed to rigorously assess LVLMs performance across diverse multimodal mathematical reasoning tasks with explicit visual dependencies.

**Visual Mathematical Reasoning.** Mathematical reasoning is a core cognitive ability increasingly explored within the context of LVLMs research Hendrycks et al. (2021); Cobbe et al. (2021b). While earlier benchmarks such as GSM8K Cobbe et al. (2021b) and MATH Hendrycks et al. (2021) primarily focused on text-based mathematical problems, recent research has expanded toward visual mathematical reasoning, incorporating diagrams, charts, and geometry-based problem-solving Wang et al. (2024b); Yang et al. (2024).

Multimodal mathematical reasoning requires LVLMs to integrate visual perception and logical reasoning, presenting a greater challenge compared to purely textual problems. Recent benchmarks like MathVista Lu et al. (2023) and MathGLM-Vision Yang et al. (2024) have advanced evaluation efforts but still suffer from issues including ambiguous annotations, dependency on GPT-based scoring methods, and limited evaluation of generalizable cognitive abilities Yan et al. (2024).

To overcome these challenges, we proposeVCBENCH, a comprehensive benchmark explicitly designed for multimodal mathematical reasoning with visual dependencies. VCBENCH encompasses 17 distinct subtasks, systematically assessing foundational cognitive skills such as temporal reasoning, logical reasoning, spatial reasoning, geometric reasoning, and pattern recognition. By standardizing task instructions and employing a multiple-choice evaluation format, VCBENCH provides objective, reproducible evaluations, offering deeper insights into the strengths and limitations of current LVLMs.

## 3 VCBENCH

### 3.1 BENCHMARK CONSTRUCTION

For VCBENCH, we employed a systematic approach to collect high-quality multimodal mathematical reasoning problems that explicitly require visual reasoning. We started by examining elementary school mathematics online question banks, manually filtering for problems that contained at least two images. In our manual review, we further excluded problems that contained non-English annotations not inferable from visual cues, were in multiple-choice format, had low-resolution or unclear visuals, relied on region-specific or cultural knowledge, or were ambiguous or not confidently understood by the reviewers. The benchmark prioritizes vision-centric evaluation through perceptual reasoning tasks that avoid specialized knowledge, while simultaneously challenging models to implicitly integrate and synthesize visual information across multiple images - a critical capability for real-world applications where understanding emerges from connecting disparate visual cues. During our selection process, we enforced strict criteria to ensure quality and consistency. We only retained problems with unique, unambiguous answers to facilitate objective evaluation. After the initial collection phase, we utilized large language models to translate all problems into English (the specific prompts used

are available in the Appendix), followed by rigorous human verification to maintain translation accuracy. The human verification process served as a filtering mechanism, where we eliminated problems containing non-English content in images, as well as those with unclear visual elements or ambiguous instructions. This meticulous curation process ensured that our benchmark evaluates genuine reasoning abilities rather than testing models on their capacity to handle poorly defined problems. Through this methodology, we assembled our final collection of problems that encompass various mathematical domains while maintaining consistent quality standards.

## 3.2 BENCHMARK STATISTICS

VCBENCH comprises a diverse collection of multimodal mathematical reasoning problems, carefully organized into **six** major categories to provide comprehensive coverage of different cognitive dimensions. As shown in Table 1, our benchmark contains 1,720 question-answer pairs featuring a total of 6,697 images. Each question is paired with, on average, 3.9 images; some problems are highly complex and include as many as 18 images, while the minimum per question is 2 images. To systematically evaluate the breadth of reasoning skills, we classified our problems into six major domains, each capturing distinct aspects of mathematical cognition. This domain-specific organization enables a granular assessment of model performance across a diverse set of cognitive abilities, ranging from visual perception and spatial understanding to arithmetic and logical reasoning. Such structured categorization not only facilitates targeted diagnostics of model strengths and weaknesses, but also mirrors the multifaceted nature of human mathematical problem-solving. Furthermore, we deliberately constrained the vocabulary used in VCBench to 2,312 unique words, minimizing confounding effects from linguistic complexity and ensuring that evaluation focuses squarely on reasoning capability. With an average question length of 136.2 characters, each problem remains concise, yet provides sufficient detail and context to support an accurate solution. The six domains are:

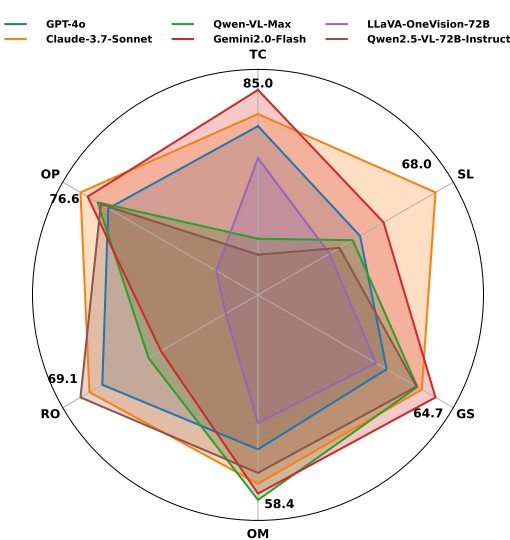

Figure 4: Comparative performance (%) of six various prominent LVLMs across six categories: Time and Calendar (TC), Space and Location (SL), Geometry and Shapes (GS), Objects and Motion (OM), Reasoning and Observation (RO), and Organization and Pattern (OP).

- **Time and Calendar**: Problems testing temporal reasoning across two subcategories (*Calendar* and *Clock*) that require understanding time intervals, and calendar-based calculations.

- **Space and Location**: Challenges focused on spatial reasoning (*Direction*, *Location*, and *Place*) that assess understanding of relative positions, directions, and spatial relationships.

- **Geometry and Shapes**: Problems spanning five subcategories (*Angle*, *Quad*, *Rectangular*, *Shape*, and *Triangle*) that test fundamental geometric comprehension from basic shape recognition to more complex property analysis.

- **Objects and Motion**: Tasks in two subcategories (*Cube* and *Move*) that evaluate the understanding of three-dimensional objects and motion transformations.

- **Reasoning and Observation**: Problems in two subcategories (*Reasoning* and *Observe*) designed to test logical reasoning and careful visual observation skills.

- **Organization and Pattern**: Challenges across three subcategories (*Organize*, *Pattern*, and *Weight*) that assess pattern recognition, sequencing, and organizational logic.

Table 2: Performance of various vision-language models (Close-Source, Open-Source, and Math Specialist categories) on a Multi-image setting across multiple tasks, including Time and Calendar, Space and Location, Geometry and Shapes, Objects and Motion, Reasoning and Observation, and Organization and Pattern.

| Models | Time and Calendar | | Space and Location | | | | Geometry and Shapes | | | | Objects and Motion | | Reasoning and Observation | | Organization and Pattern | | | Avg. |
|---|---|---|---|---|---|---|---|---|---|---|---|---|---|---|---|---|---|---|
| | Calender | Clock | Direction | Location | Place | Angle | Quad | Rectangular | Shape | Triangle | Cube | Move | Reasoning | Observe | Organize | Pattern | Weight | |
| Random Guess | 33.33 | 32.78 | 25.00 | 29.81 | 33.33 | 31.00 | 27.63 | 29.17 | 31.84 | 29.01 | 28.37 | 29.35 | 33.33 | 29.41 | 30.17 | 31.32 | 33.33 | 29.83 |
| Human | 100.00 | 96.00 | 100.00 | 93.85 | 96.67 | 95.60 | 96.84 | 95.00 | 94.02 | 94.07 | 97.67 | 94.63 | 100.00 | 93.59 | 93.20 | 95.52 | 100.00 | 93.30 |
| *Close-Source Models* | | | | | | | | | | | | | | | | | | |
| GPT-4o-mini OpenAI et al. (2024) | 80.00 | 60.66 | 0.00 | 38.46 | 53.33 | 38.40 | 21.05 | 53.57 | 37.99 | 55.56 | 32.19 | 38.24 | 0.00 | 28.68 | 60.00 | 41.38 | 100.00 | 34.88 |
| GPT-4o OpenAI et al. (2024) | 100.00 | 40.00 | 20.00 | 30.77 | 66.67 | 46.00 | 57.89 | 28.57 | 50.22 | 51.85 | 37.67 | 50.37 | 90.00 | 31.27 | 76.00 | 37.93 | 80.00 | 40.29 |
| Claude-3.7-Sonnet Anthropic (2025) | 100.00 | 50.00 | 100.00 | 53.85 | 50.00 | 58.00 | 63.16 | 57.14 | 60.70 | 59.26 | 40.41 | 67.28 | 100.00 | 31.27 | 76.40 | 53.45 | 100.00 | 46.63 |
| Qwen-VL-Max Bai et al. (2023) | 0.00 | 46.67 | 0.00 | 42.31 | 66.67 | 74.00 | 52.63 | 53.57 | 54.15 | 66.67 | 56.16 | 60.66 | 50.00 | 35.27 | 68.00 | 39.66 | 100.00 | 47.03 |
| Gemini2.0-Flash Deepmind (2025) | 100.00 | 70.00 | 20.00 | 57.69 | 66.67 | 70.00 | 68.42 | 53.57 | 61.14 | 70.37 | 44.52 | 68.75 | 40.00 | 35.53 | 74.00 | 46.55 | 100.00 | 49.77 |
| *Open-Source Models* | | | | | | | | | | | | | | | | | | |
| Emu2-Chat Sun et al. (2024b) | 0.00 | 13.33 | 0.00 | 3.85 | 0.00 | 4.00 | 10.53 | 10.71 | 12.66 | 3.70 | 8.90 | 6.99 | 0.00 | 3.62 | 0.00 | 3.45 | 20.00 | 6.05 |
| Idefics3-8B Laurençon et al. (2024) | 0.00 | 3.33 | 20.00 | 15.38 | 33.33 | 11.60 | 10.53 | 17.86 | 23.14 | 3.70 | 9.59 | 16.91 | 0.00 | 9.69 | 8.40 | 15.52 | 0.00 | 12.91 |
| DeepSeek-VL2 Wu et al. (2024) | 0.00 | 23.33 | 0.00 | 23.08 | 16.67 | 14.00 | 10.53 | 14.29 | 29.69 | 14.81 | 6.85 | 18.38 | 10.00 | 9.43 | 44.00 | 20.69 | 0.00 | 15.47 |
| Phi-3.5-vision-instruct Abdin et al. (2024) | 0.00 | 23.33 | 100.00 | 19.23 | 66.67 | 16.00 | 15.79 | 28.57 | 27.07 | 33.33 | 22.60 | 22.79 | 0.00 | 21.32 | 34.00 | 12.07 | 0.00 | 22.73 |
| InternVL2.5-8B Chen et al. (2024) | 0.00 | 33.33 | 0.00 | 34.62 | 50.00 | 34.00 | 31.58 | 50.00 | 35.81 | 37.04 | 23.29 | 25.74 | 0.00 | 18.99 | 38.00 | 6.90 | 0.00 | 24.71 |
| Llama-3.2-90B-Vision-Instruct AI (2024) | 20.00 | 24.67 | 100.00 | 11.54 | 16.67 | 26.40 | 31.58 | 32.14 | 26.20 | 22.22 | 27.40 | 25.37 | 0.00 | 25.58 | 12.00 | 29.31 | 20.00 | 25.41 |
| Qwen2.5-VL-7B-Instruct Bai et al. (2025) | 100.00 | 13.33 | 0.00 | 19.23 | 50.00 | 20.00 | 31.58 | 25.00 | 30.13 | 51.85 | 32.19 | 40.81 | 0.00 | 25.19 | 30.00 | 27.59 | 0.00 | 29.24 |
| Mantis-CLIP Jiang et al. (2024) | 0.00 | 30.00 | 80.00 | 50.00 | 66.67 | 14.00 | 15.79 | 35.71 | 38.43 | 37.04 | 19.86 | 32.35 | 40.00 | 28.04 | 52.40 | 22.41 | 100.00 | 30.23 |
| Mistral-Small-3.1-24B-Instruct Mistral (2025) | 20.00 | 40.00 | 0.00 | 30.77 | 30.00 | 38.00 | 31.58 | 35.71 | 29.26 | 51.85 | 30.82 | 31.62 | 50.00 | 29.59 | 38.00 | 34.48 | 20.00 | 31.34 |
| Kimi-VL-A3B-ThinkingTeam et al. (2025b) | 100.00 | 26.67 | 100.00 | 30.77 | 33.33 | 36.84 | 36.84 | 28.57 | 49.78 | 33.33 | 30.14 | 41.91 | 0.00 | 25.32 | 68.00 | 27.59 | 100.00 | 34.13 |
| LLaVA-Interleave-7B Li et al. (2024b) | 0.00 | 36.67 | 20.00 | 19.23 | 83.33 | 46.00 | 26.32 | 57.14 | 39.74 | 29.63 | 30.82 | 33.46 | 50.00 | 33.46 | 62.00 | 31.03 | 100.00 | 35.47 |
| LLaVA-OneVision-7B Li et al. (2024a) | 0.00 | 40.00 | 0.00 | 11.54 | 83.33 | 44.00 | 36.84 | 32.14 | 37.99 | 48.15 | 30.82 | 46.69 | 50.00 | 32.56 | 58.00 | 29.31 | 100.00 | 36.63 |
| Kimi-VL-A3B-InstructTeam et al. (2025b) | 0.00 | 46.67 | 0.00 | 30.77 | 83.33 | 44.00 | 47.37 | 39.29 | 43.23 | 33.33 | 34.93 | 44.49 | 50.00 | 31.31 | 58.00 | 36.21 | 0.00 | 37.33 |
| InternVL2.5-78B Chen et al. (2024) | 20.00 | 31.33 | 100.00 | 42.31 | 66.67 | 54.00 | 47.37 | 46.43 | 53.28 | 55.56 | 33.56 | 40.44 | 50.00 | 28.04 | 76.00 | 31.03 | 100.00 | 37.56 |
| Gemma3-27B-it Team et al. (2025a) | 100.00 | 50.00 | 0.00 | 38.46 | 83.33 | 48.40 | 31.58 | 32.83 | 44.07 | 40.74 | 32.83 | 47.79 | 50.00 | 32.82 | 54.00 | 31.03 | 80.00 | 38.02 |
| QVQ-72B-Preview Team (2024) | 100.00 | 43.33 | 0.00 | 46.15 | 83.33 | 58.00 | 42.11 | 46.43 | 44.10 | 62.96 | 36.30 | 48.16 | 50.00 | 28.55 | 78.00 | 48.28 | 100.00 | 39.13 |
| LLaVA-OneVision-72B Li et al. (2024a) | 0.00 | 33.33 | 0.00 | 26.92 | 66.67 | 61.20 | 57.89 | 57.14 | 60.70 | 51.85 | 41.10 | 60.29 | 100.00 | 38.24 | 82.00 | 41.38 | 80.00 | 47.67 |
| Qwen2.5-VL-72B-Instruct Bai et al. (2025) | 0.00 | 40.67 | 0.00 | 53.85 | 50.00 | 68.00 | 68.42 | 53.57 | 55.02 | 74.07 | 58.22 | 60.66 | 60.00 | 35.53 | 76.00 | 43.10 | 100.00 | 48.08 |

# 4 EXPERIMENT

## 4.1 MAIN RESULTS

There are a total of 17 subtasks for the evaluation from the perspectives of Temporal Reasoning, Spatial Reasoning, Geometric Reasoning, Logical Reasoning, and Pattern Recognition abilities over 21 VLMs. Table 2 provides detailed evaluation results across six visual reasoning tasks. Human performance is near-perfect with an average score of 93.30, while random guessing achieves only 29.83, which emphasizes that these tasks, though inherently solvable by humans, pose substantial challenges to current AI systems.

Figure 4 shows the comparative performance of six various prominent LVLMs across six tasks. Their relative strengths lie particularly in tasks requiring spatial reasoning and observational interpretation, suggesting these models have better internal representations or more effective cross-modal alignment between visual and linguistic information. However, despite these advancements, even these top-performing closed-source models exhibit notable shortcomings relative to humans, particularly in high-complexity reasoning scenarios (e.g., Geometry and Objects and Motion), reflecting an ongoing gap in advanced spatial reasoning, logical reasoning and pattern recognition capabilities.

Open-source models present an even more heterogeneous and generally lower performance landscape, indicative of diverse model architectures, varying degrees of multi-modal integration sophistication, and potentially inconsistent data quality or quantity during training. For example, large open-source models, including Qwen2.5-VL-72B-Instruct (48.08%) and LLaVA-OneVision-72B (47.67%), demonstrate performance comparable to mid-tier closed-source models. Their comparatively stronger results, particularly in Geometry and Shapes and Organization and Pattern tasks, suggest these models benefit from scale and possibly more sophisticated visual encoders or pre-training strategies. However, they still encounter substantial difficulties in tasks requiring nuanced observation or reasoning about motion and object interactions, highlighting remaining challenges in achieving cognative visual reasoning. The variability across different tasks, especially pronounced in Objects and Motion and Reasoning and Observation categories, points toward crucial areas requiring further research: enhancing temporal reasoning, improving dynamic visual understanding, and strengthening the integration of geometric and spatial cognition into visual-language models.

## 4.2 EVALUATION IN SINGLE-IMAGE SETTING

The evaluation is also conducted in a single-image setting for comparison. In single-image setting, we integrate visual and textual elements into a cohesive layout as shown in Figure 3. If a model performs well in single-image but poorly in multi-image, it suggests the model lacks compositional reasoning ability to link separate inputs.

The results in Table 3 reveal two key findings: First, most models perform significantly better in single-image settings compared to multi-image scenarios (average improvement of +42.3%), indicating a

Table 3: Performance comparison of vision-language models across different categories in single-image settings. The rightmost column shows the performance improvement ratio when switching from multi-image to single-image settings.

| Models | Time and Calendar | | Space and Location | | | Angle | Geometry and Shapes | | | | Objects and Motion | | Reasoning and Observation | | Organization and Pattern | | | Avg. | Improvement Ratio |
|---|---|---|---|---|---|---|---|---|---|---|---|---|---|---|---|---|---|---|---|
| | Calender | Clock | Direction | Location | Place | | Quad | Rectangular | Shape | Triangle | Cube | Move | Reasoning | Observe | Organize | Pattern | Weight | | |
| Random Guess | 33.33 | 32.78 | 25.00 | 33.33 | 31.00 | 27.63 | 29.17 | 31.84 | 29.01 | 29.35 | 28.37 | 29.35 | 33.33 | 29.41 | 30.17 | 31.32 | 33.33 | 29.83 | - |
| Human | 100.00 | 96.00 | 100.00 | 93.85 | 96.67 | 95.60 | 96.84 | 95.00 | 94.02 | 94.07 | 97.67 | 94.63 | 100.00 | 93.59 | 93.20 | 95.52 | 100.00 | 93.30 | - |
| *Close-Source Models* | | | | | | | | | | | | | | | | | | | |
| GPT-4o-mini OpenAI et al. (2024) | 100.00 | 20.00 | 0.00 | 30.77 | 100.00 | 42.80 | 26.32 | 50.00 | 56.77 | 40.74 | 34.93 | 43.01 | 90.00 | 32.43 | 72.00 | 37.93 | 60.00 | 39.65 | 13.7% |
| GPT-4o OpenAI et al. (2024) | 80.00 | 40.67 | 100.00 | 42.31 | 66.67 | 68.40 | 57.89 | 64.29 | 68.12 | 44.44 | 42.47 | 56.99 | 60.00 | 30.10 | 90.40 | 44.83 | 100.00 | 45.52 | 12.9% |
| Claude-3.7-Sonnet Anthropic (2025) | 100.00 | 54.67 | 80.00 | 65.38 | 83.33 | 61.20 | 68.42 | 78.57 | 68.56 | 77.78 | 43.84 | 69.12 | 100.00 | 34.37 | 92.00 | 63.79 | 100.00 | 51.69 | 10.8% |
| Gemini2.0-Flash Deepmind (2025) | 20.00 | 66.67 | 100.00 | 61.54 | 83.33 | 58.00 | 63.16 | 42.86 | 71.62 | 59.26 | 46.58 | 73.90 | 100.00 | 39.41 | 90.00 | 46.55 | 100.00 | 53.90 | 8.3% |
| Qwen-VL-Max Bai et al. (2023) | 0.00 | 53.33 | 100.00 | 73.08 | 83.33 | 80.00 | 52.63 | 75.00 | 69.87 | 66.67 | 57.53 | 72.43 | 100.00 | 43.54 | 91.60 | 41.38 | 80.00 | 57.03 | 21.3% |
| *Open-Source Models* | | | | | | | | | | | | | | | | | | | |
| Idefics3-8B Laurençon et al. (2024) | 0.00 | 10.00 | 20.00 | 11.54 | 16.67 | 10.00 | 5.26 | 32.14 | 20.52 | 7.41 | 17.12 | 18.01 | 0.00 | 12.53 | 30.00 | 20.69 | 0.00 | 15.64 | 21.2% |
| LLaMA-3.2-90B-Vision-Instruct AI (2024) | 80.00 | 30.00 | 0.00 | 15.38 | 33.33 | 26.00 | 15.79 | 25.00 | 17.03 | 33.33 | 27.40 | 26.47 | 100.00 | 19.64 | 49.60 | 12.07 | 0.00 | 22.38 | -11.9% |
| Emu2-Chat Sun et al. (2024b) | 60.00 | 12.67 | 100.00 | 23.08 | 16.67 | 24.00 | 42.11 | 28.57 | 24.02 | 18.52 | 22.60 | 24.63 | 0.00 | 22.87 | 12.00 | 22.41 | 0.00 | 23.08 | 281.5% |
| DeepSeek-VL2 Wu et al. (2024) | 20.00 | 33.33 | 0.00 | 19.23 | 33.33 | 28.00 | 10.53 | 32.14 | 32.31 | 25.93 | 13.70 | 32.35 | 0.00 | 20.03 | 46.00 | 27.59 | 100.00 | 24.77 | 60.1% |
| Mantis-CLIP Jiang et al. (2024) | 0.00 | 35.33 | 80.00 | 23.08 | 0.00 | 28.00 | 42.11 | 46.43 | 31.88 | 11.11 | 26.03 | 25.00 | 0.00 | 27.52 | 12.00 | 34.48 | 0.00 | 27.50 | -9.0% |
| LLaVA-Interleave-7BLi et al. (2024b) | 00.00 | 30.00 | 100.00 | 30.77 | 0.00 | 26.00 | 36.84 | 42.86 | 33.19 | 14.81 | 31.51 | 26.47 | 50.00 | 29.07 | 28.00 | 25.86 | 0.00 | 29.24 | -17.6% |
| Phi-3.5-vision-instruct Abdin et al. (2024) | 0.00 | 13.33 | 80.00 | 19.23 | 16.67 | 24.00 | 10.53 | 42.86 | 34.50 | 22.22 | 32.19 | 29.78 | 20.00 | 31.40 | 46.00 | 25.86 | 100.00 | 30.93 | 36.1% |
| LLaVA-OneVision-7B Li et al. (2024a) | 0.00 | 43.33 | 0.00 | 23.08 | 100.00 | 44.00 | 21.05 | 35.71 | 44.10 | 44.44 | 30.82 | 42.65 | 40.00 | 29.07 | 64.40 | 27.59 | 80.00 | 35.47 | -3.2% |
| InternVL2.5-8B Chen et al. (2024) | 0.00 | 33.33 | 0.00 | 26.92 | 50.00 | 46.40 | 31.58 | 39.29 | 51.53 | 48.15 | 31.51 | 42.65 | 30.00 | 28.42 | 60.80 | 29.31 | 80.00 | 36.16 | 46.3% |
| Gemma3-27B-itTeam et al. (2025a) | 80.00 | 40.00 | 0.00 | 26.92 | 33.33 | 48.40 | 21.05 | 57.14 | 45.85 | 33.33 | 33.56 | 45.22 | 100.00 | 30.10 | 66.80 | 20.69 | 60.00 | 36.80 | 2.1% |
| Kimi-VL-A3B-ThinkingTeam et al. (2025b) | 0.00 | 33.33 | 0.00 | 34.62 | 50.00 | 62.00 | 52.63 | 39.29 | 52.40 | 77.78 | 26.03 | 55.15 | 50.00 | 25.19 | 86.00 | 39.66 | 100.00 | 38.72 | 13.4% |
| LLaVA-OneVision-72B Li et al. (2024a) | 20.00 | 53.33 | 0.00 | 30.77 | 33.33 | 38.00 | 47.37 | 39.29 | 51.53 | 55.56 | 39.73 | 41.54 | 100.00 | 32.95 | 32.00 | 55.17 | 100.00 | 39.24 | -17.7% |
| Mistral-Small-3.1-24B-Instruct Mistral (2025) | 20.00 | 40.00 | 0.00 | 38.46 | 50.00 | 64.00 | 57.89 | 46.43 | 56.77 | 70.37 | 30.14 | 50.74 | 100.00 | 31.65 | 82.00 | 43.10 | 80.00 | 42.21 | 34.7% |
| QVQ-72B-Preview Team (2024) | 80.00 | 41.33 | 80.00 | 61.54 | 50.00 | 64.00 | 68.42 | 39.29 | 58.95 | 81.48 | 32.19 | 64.34 | 50.00 | 35.01 | 90.00 | 50.00 | 100.00 | 47.44 | 21.2% |
| InternVL2.5-78B Chen et al. (2024) | 80.00 | 50.00 | 100.00 | 42.31 | 50.00 | 62.80 | 63.16 | 57.14 | 65.94 | 55.56 | 32.19 | 61.76 | 90.00 | 36.43 | 88.00 | 36.21 | 100.00 | 47.73 | 27.1% |
| Kimi-VL-A3B-InstructTeam et al. (2025b) | 0.00 | 70.00 | 100.00 | 50.00 | 66.67 | 50.00 | 31.58 | 35.71 | 59.39 | 51.85 | 46.58 | 62.13 | 50.00 | 38.11 | 82.00 | 46.55 | 100.00 | 48.37 | 29.6% |
| Qwen2.5-VL-7B-Instruct Bai et al. (2025) | 0.00 | 53.33 | 100.00 | 46.15 | 83.33 | 72.80 | 52.63 | 60.71 | 61.14 | 55.56 | 60.96 | 64.34 | 100.00 | 37.86 | 92.00 | 36.21 | 80.00 | 51.10 | 74.8% |
| Qwen2.5-VL-72B-Instruct Bai et al. (2025) | 20.00 | 55.33 | 100.00 | 73.08 | 83.33 | 80.00 | 52.63 | 75.00 | 69.87 | 66.67 | 57.53 | 72.43 | 90.00 | 43.54 | 92.00 | 41.38 | 100.00 | 57.03 | 18.6% |
| *Math Specialist Models* | | | | | | | | | | | | | | | | | | | |
| G-LLaVA-13B Gao et al. (2023) | 0.00 | 40.00 | 0.00 | 23.08 | 33.33 | 20.40 | 31.58 | 32.14 | 26.64 | 25.93 | 15.75 | 26.10 | 0.00 | 26.49 | 24.00 | 24.14 | 20.00 | 25.47 | - |
| G-LLaVA-7B Gao et al. (2023) | 100.00 | 36.67 | 20.00 | 30.77 | 0.00 | 32.00 | 21.05 | 50.00 | 31.88 | 40.74 | 23.97 | 27.21 | 0.00 | 27.26 | 28.00 | 24.14 | 100.00 | 28.26 | - |
| MathLlava Shi et al. (2024) | 100.00 | 20.00 | 80.00 | 26.92 | 0.00 | 32.00 | 31.58 | 21.43 | 27.51 | 11.11 | 34.93 | 29.04 | 40.00 | 29.97 | 28.40 | 29.31 | 80.00 | 29.30 | - |

Table 4: Influence of Chain-of-Thought Wei et al. (2023a) on model performances.

| Model | CoT | Time and Calendar | | Space and Location | | | Angle | Geometry and Shapes | | | | Objects and Motion | | Reasoning and Observation | | Organization and Pattern | | | Avg. |
|---|---|---|---|---|---|---|---|---|---|---|---|---|---|---|---|---|---|---|---|
| | | Calender | Clock | Direction | Location | Place | | Quad | Rectangular | Shape | Triangle | Cube | Move | Reasoning | Observe | Organize | Pattern | Weight | |
| GPT-4o OpenAI et al. (2024) | ✗ | 100.00 | 40.00 | 20.00 | 30.77 | 66.67 | 46.00 | 57.89 | 28.57 | 50.22 | 51.85 | 37.67 | 50.37 | 90.00 | 31.27 | 76.00 | 37.93 | 80.00 | 40.29 |
| | ✓ | 100.00 | 40.00 | 0.00 | 38.46 | 66.67 | 52.00 | 63.16 | 32.14 | 53.71 | 66.67 | 33.56 | 52.57 | 100.00 | 30.75 | 82.00 | 58.62 | 100.00 | 42.03 |
| | | 0.00 | 0.00 | -20.00 | +7.69 | 0.00 | +6.00 | +5.27 | +3.57 | +3.49 | +14.82 | -4.11 | +2.20 | +10.00 | -0.52 | +6.00 | +20.69 | +20.00 | +1.74 |
| Qwen-VL-Max Bai et al. (2023) | ✗ | 0.00 | 46.67 | 0.00 | 42.31 | 66.67 | 74.00 | 52.63 | 42.86 | 54.15 | 66.67 | 56.16 | 60.66 | 50.00 | 35.27 | 68.00 | 39.66 | 100.00 | 47.03 |
| | ✓ | 20.00 | 36.67 | 100.00 | 57.69 | 66.67 | 74.40 | 52.63 | 57.14 | 60.26 | 77.78 | 52.74 | 61.03 | 90.00 | 36.05 | 93.60 | 44.83 | 100.00 | 49.48 |
| | | +20.00 | -10.00 | +100.00 | +15.38 | 0.00 | +0.40 | 0.00 | +14.28 | +6.11 | +11.11 | -3.42 | +0.37 | +40.00 | +0.78 | +25.60 | +5.17 | 0.00 | +2.45 |
| Gemini2.0-Flash Deepmind (2025) | ✗ | 100.00 | 70.00 | 20.00 | 57.69 | 66.67 | 70.00 | 68.42 | 53.57 | 61.14 | 70.37 | 44.52 | 68.75 | 40.00 | 35.53 | 74.00 | 46.55 | 100.00 | 49.77 |
| | ✓ | 80.00 | 83.33 | 20.00 | 69.23 | 83.33 | 66.40 | 68.42 | 67.86 | 71.62 | 66.67 | 41.10 | 70.96 | 100.00 | 37.86 | 89.40 | 56.90 | 100.00 | 53.66 |
| | | -20.00 | +13.33 | 0.00 | +11.54 | +16.66 | -3.60 | 0.00 | +14.29 | +10.48 | -3.70 | -3.42 | +2.21 | +60.00 | +2.33 | +15.40 | +10.35 | 0.00 | +3.89 |

strong bias toward single-image optimization. For instance, Qwen-VL-Max shows a +21.3% gain in single-image performance, while models like Emu2-Chat exhibit dramatic improvements (+281.5%). Second, specialized multi-image models like LLaVA-Interleave-7B show the opposite trend (-17.6% in single-image mode), achieving higher accuracy in multi-image tasks than in single-image ones. This contrast suggests that unlike dedicated multi-image architectures, conventional models struggle to integrate visual information across multiple inputs, highlighting a critical limitation in current vision-language systems. Addressing this gap by effectively leverage cross-image cues for reasoning remains an essential challenge for future research.

## 4.3 RESULTS OF MATH SPECIALIST MODELS

The Math Specialist models, including G-LLaVA-13B, G-LLaVA-7B, and MathLlava, exhibit relatively low overall performance, with average scores from 25.47 to 29.30. Notably, G-LLaVA-13B records the lowest score at 25.47, while MathLlava achieves a slightly higher score of 29.30. Though these models are designed to focus on mathematical reasoning, their performance across diverse tasks-such as time and calendar, spatial reasoning, and geometric challenges-remains inconsistent. For example, while G-LLaVA-7B reaches a perfect score (100.00) on the Calendar sub-task, its scores in other categories, such as Clock and geometry-related tasks, are considerably lower.

Furthermore, the results indicate that these Math Specialist models struggle to match the performance of their general-purpose counterparts. Despite showing some strengths-for example, MathLlava scoring 34.93 on the Cube task-these models fall short on several key aspects, including Clock, Location, and reasoning tasks. This pattern underscores the challenge of integrating specialized mathematical capabilities with the broader spectrum of visual understanding.

## 5 ANALYSIS

### 5.1 INFLUENCE OF CHAIN-OF-THOUGHT ON MODEL PERFORMANCE

Chain-of-thought Wei et al. (2023a) reasoning generally enhances model performance, as the Table 4 shows stable improvements across several domains when CoT is enabled. For instance, Qwen-VL-Max exhibits a dramatic 40% boost in the "Reasoning" task, highlighting the significant impact of structured reasoning on spatial understanding. Gemini2.0-Flash also benefits substantially, with a

Table 5: Comparisons between existing visual math benchmarks for LVLMs.

| Benchmark | Image Numbers | Question Numbers | Required Skills | | | | | Multi-Images | Answer Type |
|---|---|---|---|---|---|---|---|---|---|
| | | | Temporal | Spatial | Geometric | Logical | Pattern | | |
| Olympiadbench He et al. (2024) | 5,129 | 8,952 | ✗ | ✗ | ✓ | ✗ | ✗ | ✗ | Free-form |
| GeoQA Chen et al. (2021) | 4,998 | 4,998 | ✗ | ✗ | ✓ | ✗ | ✗ | ✗ | Multiple Choice |
| MATH-Vision Wang et al. (2024a) | 3,472 | 3,040 | ✗ | ✗ | ✓ | ✗ | ✗ | ✓ | Free-form & Multiple Choice |
| MathVista Lu et al. (2023) | 5,487 | 6,141 | ✓ | ✗ | ✓ | ✓ | ✗ | ✓ | Free-form & Multiple Choice |
| MMMU$_{math}$ Yue et al. (2024) | 577 | 540 | ✗ | ✗ | ✓ | ✗ | ✗ | ✗ | Free-form & Multiple Choice |
| GeoMath Xu et al. (2024) | 4,540 | 9,155 | ✗ | ✗ | ✓ | ✓ | ✗ | ✗ | Free-form & Multiple Choice & Prove |
| U-Math Chernyshev et al. (2025) | 225 | 1,100 | ✗ | ✗ | ✓ | ✗ | ✗ | ✓ | Free-form |
| Blink Fu et al. (2024) | 7,358 | 3,807 | ✗ | ✗ | ✗ | ✗ | ✗ | ✓ | Multiple Choice |
| MM-MATH Sun et al. (2024a) | 5,929 | 5,929 | ✗ | ✗ | ✓ | ✗ | ✗ | ✗ | Free-form |
| MMIE$_{math}$ Xia et al. (2024) | 26,534 | 20,103 | ✗ | ✗ | ✓ | ✗ | ✗ | ✓ | Free-form & Multiple Choice |
| Polymath Gupta et al. (2024) | 5,000 | 5,000 | ✗ | ✗ | ✓ | ✗ | ✓ | ✗ | Multiple Choice |
| NTSEBench Pandya et al. (2025) | 4,642 | 2,728 | ✗ | ✓ | ✗ | ✓ | ✓ | ✓ | Multiple Choice |
| BSA [1] Xu et al. (2025) | 312 | 312 | ✗ | ✓ | ✗ | ✗ | ✗ | ✓ | Multiple Choice |
| MV-MATH Wang et al. (2025) | 6,061 | 2,009 | ✗ | ✓ | ✓ | ✓ | ✓ | ✓ | Free-form & Multiple Choice |
| **Ours** | 6,697 | 1,720 | ✓ | ✓ | ✓ | ✓ | ✓ | ✓ | Multiple Choice |

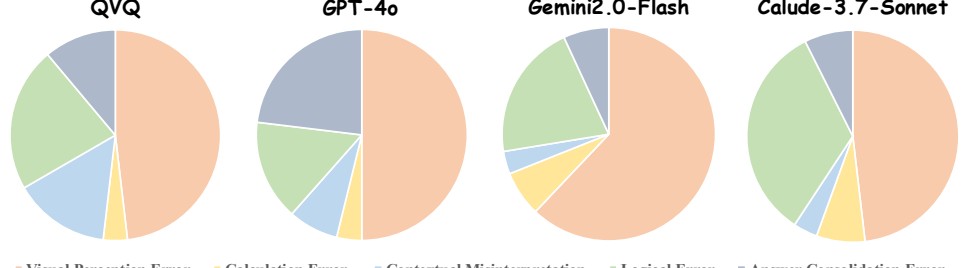

Figure 5: A comparison of error distributions among three model, GPT-4o, Gemini2.0-Flash, and Calude-3.7-Sonnet, across five error categories: visual perception errors, calculation errors, contextual misunderstandings, logical errors, and answer integration errors.

15.40 point increase in the "Pattern" category and a 16.66 point rise in "Place" suggesting that CoT particularly aids in tasks requiring complex organizational and geometric reasoning.

While improvements are evident, the efficacy of chain-of-thought (CoT) prompting exhibits strong task-dependent variation. CoT consistently enhances performance in multi-step reasoning tasks (e.g., Pattern and Reasoning tasks), where all models show gains. However, it proves neutral in perception-heavy tasks (e.g., Calender and Direction tasks) due to interference with low-level spatial or temporal processing. Nonetheless, the overall trend supports that incorporating CoT tends to enhance problem-solving abilities, especially in tasks that demand high-level reasoning and pattern recognition.

## 5.2 COMPARISON WITH OTHER BENCHMARKS

In comparison to existing visual math benchmarks, our dataset stands out in several important ways as shown in Table 5. While benchmarks such as Olympiadbench He et al. (2024) and GeoQA Chen et al. (2021) focus primarily on specific skills like geometry and logical reasoning, our benchmark includes a broader spectrum of required skills, including temporal, spatial, geometric, logical, and pattern recognition. This comprehensive skill coverage provides a more holistic evaluation of LVLMs. Additionally, our dataset supports multi-image tasks, a feature not widely supported by other benchmarks such as Blink Fu et al. (2024) and GeoQA Chen et al. (2021), enhancing its applicability for real-world tasks that require understanding across multiple visual inputs. Moreover, our benchmark boasts a higher image-question ratio than other benchmarks, meaning that on average, each question is associated with more images. Finally, our dataset offers multiple-choice answer types for easier evaluation, unlike other benchmarks that provide free-form answer format which is hard to evaluate, such as MM-MATH Sun et al. (2024a) and U-Math Chernyshev et al. (2025).

## 5.3 ERROR DISTRIBUTION FOR VCBENCH

We define five error types in this benchmark: Visual Perception Error indicates that the model misinterprets or fails to accurately perceive visual content; Calculation Error captures mistakes made during arithmetic computations; Contextual Misinterpretation occurs when the model misreads the textual conditions, such as treating unrelated information as relevant; Logical Error refers to flaws in the reasoning process; and Answer Consolidation Error encompasses failures to directly answer the question or instances where multiple, conflicting answers are provided. We conduct manual

error classification for all questions across four top-tier models, enabling precise identification of each model's failure patterns and relative weaknesses across different error categories. As shown in Figure 5, Visual Perception Errors are predominant across all models, with Gemini2-Flash exhibiting the highest rate at about 62%. This persistent pattern across architectures suggests that enhancing visual perception capabilities remains the most critical challenge for multimodal models. Calculation Errors remain consistently low (ranging from about 4% to about 7%), indicating that basic arithmetic computation has become relatively robust in modern models. Contextual Misinterpretation errors are minimal, particularly for Gemini2-Flash (about 3%) and Claude (about 4%), which indicates a relatively robust understanding of textual context. However, QVQ's comparatively higher rate (6%) may reflect its tendency toward over-reasoning, where excessive analysis leads to detachment from the original question context.

On the other hand, discrepancies are more apparent in the Logical and Answer Consolidation Error rates. Claude shows a significantly high Logical Error rate of about 33% compared to GPT-4o's about 15% and QVQ's about 22%, revealing the weaknesses in its deductive reasoning pipelines. Moreover, while Answer Consolidation Errors are generally low (QVQ at about 11% and both Gemini2-Flash and Claude at about 7%), GPT-4o presents a higher rate of about 23%, suggesting its advanced reasoning capabilities may come at the cost of response discipline, where the model sometimes generates multiple answers rather than a single one. This trade-off between exploratory reasoning and answer precision presents an important optimization target for future iterations.

## 5.4 ANALYSIS OF PROBLEM DIFFICULTY AND MODEL PERFORMANCE

All questions in our benchmark were sourced from established online question banks and annotated by editors with a difficulty coefficient ranging from 0.0 to 1.0. Questions with coefficients between 0.0 and 0.35 are categorized as easy, those between 0.35 and 0.75 as medium, and those from 0.75 to 1.0 as hard. Overall, 27.7% of questions are classified as easy, 41.6% as medium, and 30.7% as hard. Interestingly, the results in Table 6 reveal that questions annotated as hard tend to yield higher accuracy,

Table 6: Accuracy comparison of various models on questions categorized by difficulty along with their average performance.

| Models | Easy | Medium | Hard | Avg. |
|---|---|---|---|---|
| LLaMA-3.2-90B-Vision-Instruct | 22.22 | 26.15 | 23.89 | 25.41 |
| Mantis-CLIP | 29.63 | 29.30 | 32.37 | 30.23 |
| InternVL2.5-78B | 25.93 | 36.03 | 41.62 | 37.56 |
| QVQ-72B-Preview | 18.52 | 36.71 | 45.66 | 39.13 |
| LLaVA-OneVision-72B | 29.63 | 45.32 | 53.76 | 47.62 |
| Qwen2.5-VL-72B-Instruct | 25.93 | 45.49 | 55.11 | 48.08 |

while the easy and medium problems register lower accuracy. This counterintuitive outcome may be attributed to the fact that simpler questions, which primarily require the identification of patterns rather than intricate computations, pose a different challenge compared to the hard questions that demand complex calculation and structured reasoning. It is important to note that the difficulty levels in our benchmark are derived from the original textbook platforms, where they are based on large-scale student performance statistics and thus reflect authentic human difficulty. Upon closer analysis, we observed that models often perform worse on problems considered "easy" by human standards. These are typically perception-heavy tasks. For instance, in clock reading questions, models struggle to recognize the positions of hour and minute hands, particularly when the clocks have decorative or irregular designs. Similarly, in block counting tasks, the presence of stacked or overlapping cubes frequently confuses model predictions due to challenges in depth perception and object segmentation. Although both types of questions are suitable for elementary students and considered trivial by humans, they expose current models' significant limitations in low-level visual understanding, leading to surprisingly low performance on these "easy" items.

## 6 CONCLUSION

This paper introduces VCBENCH—a comprehensive evaluation framework designed to assess multimodal mathematical reasoning with explicit visual dependency. By addressing the limitations of existing datasets in multi-image integration and cross-modal relational reasoning, our benchmark provides a detailed analysis of 26 state-of-the-art LVLMs across six cognitive domains and 17 task categories. The evaluation reveals significant performance disparities, particularly in areas such as multi-step instruction following, basic visual perception, cross-image consistency, and vulnerability to visual hallucinations.

## 7 ETHICS STATEMENT

This research complies with the ICLR Code of Ethics. All Large Vision-Language Models (LVLMs) evaluated in this study are publicly available and widely adopted in academia and industry. VCBENCH was constructed using synthetic and open-access visual content, without involving any personally identifiable, sensitive, or proprietary information. Our work is limited to methodological exploration and model evaluation, with no direct experimentation on human subjects or real-world deployment. While the benchmark is designed to rigorously assess multimodal mathematical reasoning, we recognize that advancements in LVLMs may inadvertently reinforce existing biases embedded in training corpora. We encourage future researchers and practitioners to systematically take fairness, bias, and robustness into account when applying or extending our methodology. The authors declare no conflicts of interest associated with this submission.

## 8 REPRODUCIBILITY STATEMENT

We have taken extensive measures to ensure the reproducibility of our findings with VCBENCH. The benchmark, including all problems, images, and evaluation protocols, is described in detail in Sections 3 and 4. We provide comprehensive documentation on dataset construction, problem generation, cognitive domain taxonomy, and visual dependency design. Evaluation procedures for all 26 LVLMs—including model configurations, scoring metrics, and inference setups—are transparently reported in the main text and further clarified in the Appendix. All preprocessing, benchmark scripts, and evaluation code will be released alongside the paper, enabling independent replication of our results. In addition, the full problem set, associated images, and baseline evaluation outputs will be released as part of the anonymous supplementary materials, to facilitate benchmarking, comparison, and future work in multimodal mathematical reasoning.

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

# A APPENDIX

## A.1 EXPERIMENT DETAILS

Table 7: Generation parameters for LVLMs (with grouped configurations).

| Model | Generation Setup |
|---|---|
| GPT-4o-mini & GPT-4o | API URL: `https://api.openai.com/v1/chat/completions` temperature = 0.2, max_tokens = 1024 |
| Claude-3.7-Sonnet | API URL: `https://api.anthropic.com/v1/messages`, temperature = 0.2, max_tokens = 1024 |
| Gemini2.0-Flash | API URL: `https://generativelanguage.googleapis.com/v1beta/models/gemini-pro:generateContent`, temperature = 0.2, max_tokens = 1024 |
| Qwen-VL-Max | Use dashscope package, temperature = 0.2, max_new_tokens = 1024 |
| Open-Source Models | *Same parameters for all below:* Deployed by vllm, with do_sample = True, temperature = 0.2, max_new_tokens = 1024 • Idefics3-8B • LLaMA-3.2-90B-Vision-Instruct • Emu2-Chat • DeepSeek-VL2 • Mantis-CLIP • LLaVA-Interleave-7B • Phi-3.5-vision-instruct • InternVL-2.5 • LLaVA-OneVision-7B/72B • Gemma3-27B-it • Mistral-Small-3.1-24B-Instruct • Qwen2.5-VL-7B/72B-Instruct |
| QVQ-72B-Preview | do_sample = True, temperature = 0.2, max_new_tokens = 2048 |
| G-LLaVA-7B/13B | do_sample = True, temperature = 0.2, max_new_tokens = 1024 |
| MathLlava | do_sample = True, temperature = 0.2, max_new_tokens = 1024 |

## A.2 IMPACT OF IMAGE COUNT ON MODEL ACCURACY

Table 8: Model accuracy (%) across problems grouped by image count. The number of questions per image count is shown in the second row. As image count increases, accuracy generally decreases with some fluctuations.

| Model / Image Count | 2 images | 3 images | 4 images | 5 images | 6–7 images | 8–10 images | 11+ images |
|---|---|---|---|---|---|---|---|
| Question Count | 316 | 544 | 515 | 221 | 76 | 34 | 14 |
| GPT-4o | 58.9 | 47.3 | 44.7 | 40.3 | 35.5 | 30.9 | 21.4 |
| Claude-3.7-sonnet | 63.5 | 54.8 | 50.1 | 46.7 | 40.8 | 37.2 | 23.1 |
| Qwen-VL-max | 61.9 | 52.1 | 48.0 | 45.2 | 39.7 | 36.7 | 18.7 |
| Gemini-2.0-flash | 65.2 | 57.1 | 53.8 | 51.2 | 44.1 | 39.8 | 26.2 |

Table 8 presents the accuracy of four leading multimodal models across problems grouped by the number of images provided in each question. We observe a general trend: as the number of images increases, average model accuracy tends to decrease. These results underscore the importance of developing models capable of robust visual reasoning, especially in scenarios requiring the integration of multiple visual inputs.

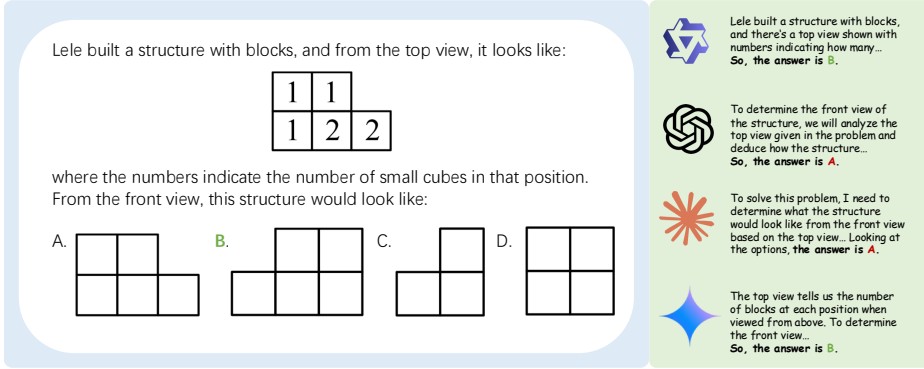

Figure 6: Case for Visual Perception Error.

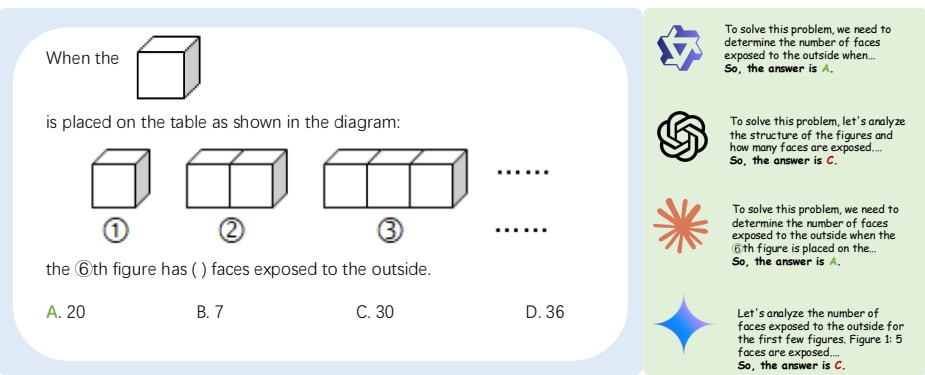

Figure 7: Case for Calculation Error.

## A.3 CASE STUDIES

Individually, each of the five cases highlights a distinct type of reasoning error made by models. In Figure 6 (Visual Perception Error), models failed to accurately map the numbers from a top view to a front view, resulting in confusion about the arrangement of cubes in three-dimensional space. Figure 7 (Calculation Error) exposed mistakes in pattern recognition and extrapolation, with models incorrectly calculating the number of exposed faces as the sequence grew. Figure 8 (Contextual Misinterpretation) showed that models struggled to correctly interpret the scenario and geometric relationships required to form a triangle, often misapplying the cutting method or misreading the intent of the task. In Figure 9 (Logical Error), models made incorrect choices about physical stability, suggesting a lack of real-world reasoning about spatial structure and balance. Figure 10 (Answer Consolidation Error) demonstrated the models' challenges in mentally rotating and consolidating three-dimensional information to select the correct two-dimensional viewpoint.

These examples reveal that current models face significant difficulties in visual and mathematical reasoning. Their limitations extend beyond simple miscalculations, reflecting deeper struggles with tasks that require the integration of visual perception, contextual understanding, logical reasoning, and spatial manipulation. In many cases, models fail to synthesize information across multiple steps or modalities, leading to errors in connecting abstract concepts to specific visual details or consolidating reasoning into correct answers. Together, these challenges suggest that current models lack the comprehensive, multi-step cognitive processing abilities required for human-level performance in complex multimodal mathematical problems.

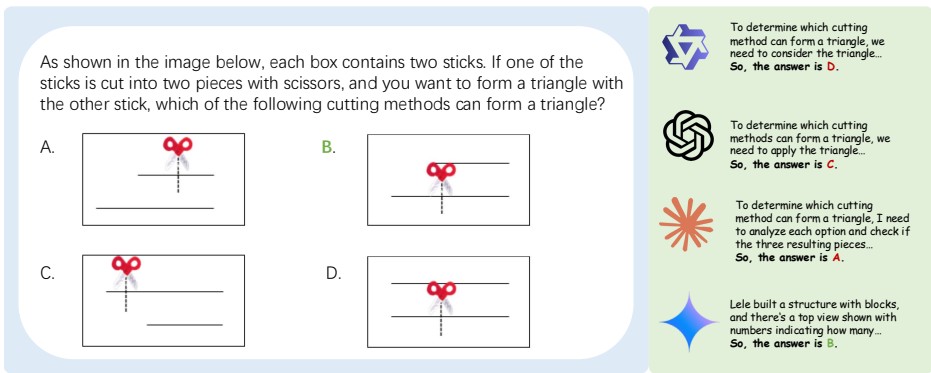

Figure 8: Case for Contextual Misinterpretation.

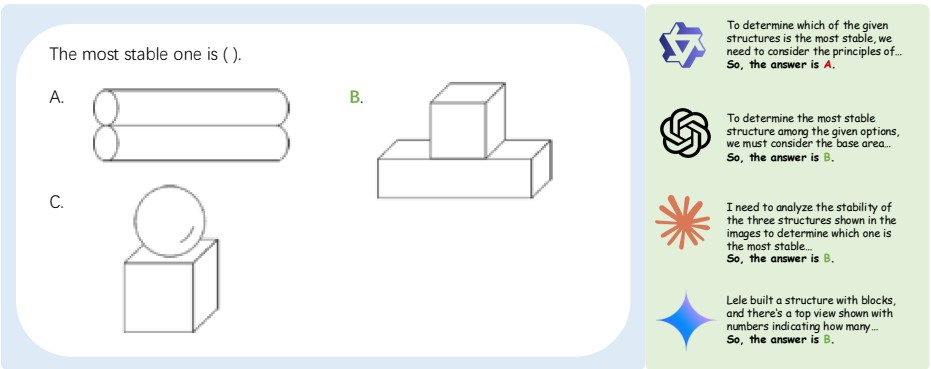

Figure 9: Case for Logical Error.

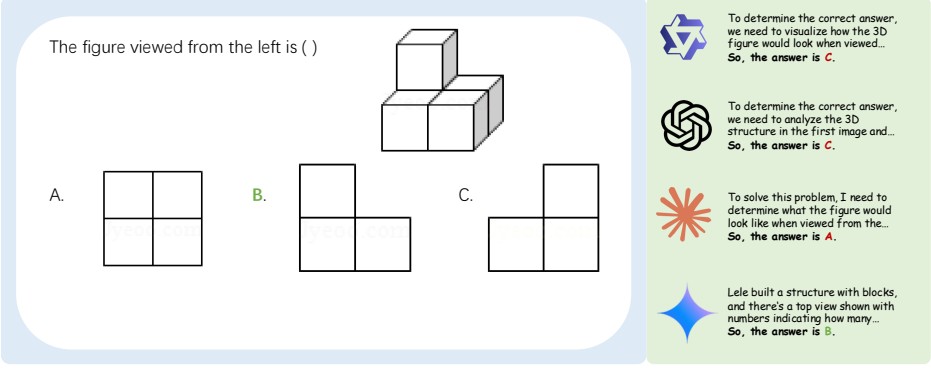

Figure 10: Case for Answer Consolidation Error.

## A.4 PROMPT FOR EXPERIMENT

Table 9: Inference Prompt.

| Inference Prompt |
| --- |
| You are a helpful AI assistant.
Please answer the following questions and output the answer options directly.
*Question: { question }* |

Table 10: Inference Prompt with Chain-of-Thought.

| Inference Prompt with Chain-of-Thought |
| --- |
| You are a helpful AI assistant.
Please think step by step before answer the following questions and the output the answer.
*Question: { question }* |

Table 11: LLM-Based Evaluation Prompt.

| LLM-Evaluation Prompt |
| --- |
| You are an answer evaluator. I will give you a response and an answer.
Please tell me whether this response is correct or wrong. Just answer **yes** or **no**.
For example,
Response: The figure that cannot be folded into a cube is: C. <image>
Correct Answer: B
So, you need to respond **no** only.
Response: The unfolded shape of the cube is: B. <image>
Correct Answer: B
So, you need to respond **yes** only.
Here is the response and correct answer I want you to evaluate.
*Response: { model response }*
*Correct Answer: { correct answer }* |

