# OpenReview forum: "Benchmarking Multimodal Mathematical Reasoning with Explicit Visual Dependency"
_ICLR.cc/2026/Conference — Submitted to ICLR 2026_

### Official Review · Reviewer_tAJE · 2025-10-28

**Soundness:** 2
**Presentation:** 3
**Contribution:** 2
**Rating:** 4
**Confidence:** 5

**Summary:**

The paper presents a math reasoning evaluation benchmark, VCBench. Specifically, its aim is to evaluate diverse VLMs on elementary school level problems where the model has to integrate information from diverse images for a single problem. The authors find that despite the easiness in the problems, the existing VLMs are not very good on this dataset. Additional analysis shows that most models are suffering from visual perception errors. Overall, the paper provides a useful resource for the community to benchmark their models. However, I believe that it lacks on several fronts in terms of its motivation, design, and experiments.

**Strengths:**

1. The paper creates a large evaluation set for assessing the capabilities of frontier VLMs on math reasoning tasks that are simpler for humans. In particular, they focus on scenarios with multiple images for a single question.
2. The paper performs a broad set of experiments to show that the existing models have a large gap with human annotators. Further, the paper shows the performance across diverse fine-grained scenarios in the dataset.
3. The work assesses the failure modes of the existing models, and finds that they suffer from perceptual errors.

**Weaknesses:**

1. I don’t agree with the paper’s motivation that existing evaluation datasets just test the knowledge-centric capabilities of VLMs. This argument is shallow without any real evidence. Many examples in MathVista and MathVision would require the model to perceive the image deeply before answering. Infact, MathVerse [1] has already shown that you can make the evaluation datasets more challenging by focusing on visual-centric question-styles.
2. The paper does not feel different from several prior works such as ReMI [2] and MuirBench [3]. Further, it is not surprising that these models are bad at understanding clocks and calendars, also highlighted in Contextual dataset [4].
3. There is hardly any information about the data acquisition procedure (Section 3.1). In an evaluation paper, the data collection process should be described in greater detail but it is just skimmed in the current version. There are many missing details: (a) which online sources are the questions taken from, (b) who are the human annotators collecting the questions, (c) how are you ensuring consistency, (d) who are the human participants solving the questions, and more.
4. There are no evaluation numbers for long CoT models like OpenAI o-series models, Gemini-Pro, Claude-Thinking etc. In fact, it is unclear whether RL-training helps in improving the performance on your dataset or not. For instance, comparison between Qwen-2.5VL and its RL-trained versions such as OpenVLThinker, VL-Rethinker, and MM-Eureka etc.

[1] MathVerse: https://arxiv.org/abs/2403.14624
[2] ReMi: https://arxiv.org/pdf/2406.09175
[3] MuirBench:  https://arxiv.org/abs/2406.09411
[4] Contextual: https://arxiv.org/abs/2401.13311

**Questions:**

Mentioned above

---

### Official Review · Reviewer_mVkB · 2025-10-30

**Soundness:** 2
**Presentation:** 2
**Contribution:** 1
**Rating:** 2
**Confidence:** 5

**Summary:**

This paper introduces VCBench, a benchmark designed to evaluate multimodal mathematical reasoning that explicitly depends on visual understanding across multiple images. The benchmark contains 1,720 problems with 6,697 images spanning six cognitive domains. It aims to test models’ abilities to integrate information across multiple visuals and textual elements.

**Strengths:**

Provides a comprehensive dataset with controlled cognitive domains and consistent annotation quality.

Offers broad model coverage (26 LVLMs), establishing a useful empirical baseline for the community.

**Weaknesses:**

The main contribution of the paper lies in dataset and benchmark construction, which represents an incremental contribution rather than a methodological advance. The work primarily offers an evaluation suite and empirical results, lacking deeper analytical or theoretical insight expected at a top-tier venue.

The proposed multi-image and visual-dependency test setting has already been explored in prior benchmarks, such as MV-MATH: Evaluating Multimodal Math Reasoning in Multi-Visual Contexts (CVPR 2025). The authors should include a more extensive literature review to contextualize their benchmark among existing multimodal mathematical reasoning datasets.

While the evaluation results are comprehensive, the paper lacks in-depth analysis or interpretation of failure cases that could provide actionable insights for future research.  Moreover, no potential solutions or architectural recommendations are discussed to address these observed limitations, which weakens the paper’s forward-looking impact.

**Questions:**

Please refer to the weakness section.

---

### Official Review · Reviewer_JRYk · 2025-10-30

**Soundness:** 3
**Presentation:** 3
**Contribution:** 3
**Rating:** 2
**Confidence:** 4

**Summary:**

This paper introduces VCBENCH, a benchmark designed to address a key deficiency in current LVLM evaluations: the neglect of foundational visual reasoning and multi-image integration capabilities. VCBENCH consists of 1,720 elementary math problems that require integrating information from multiple images to be solved. Through extensive evaluation of 26 state-of-the-art (SOTA) models, the paper reveals that even the most advanced models perform poorly , identifying visual perception and cross-image information integration as the core bottlenecks for current models.

**Strengths:**

- VCBENCH systematically makes explicit visual dependency and mandatory multi-image input the core of its evaluation.

- The experimental results demonstrate that current LVLMs perform extremely poorly on tasks requiring the integration of distributed visual information, identifying visual perception and cross-image integration as key performance bottlenecks.

- Through its division into six cognitive domains, VCBENCH not only provides an overall performance score but also enables a fine-grained analysis of a model's strengths and weaknesses across different dimensions like temporal, spatial, and geometric reasoning.

**Weaknesses:**

- The overall scale of 1,720 questions is reasonable, but when broken down into 17 subcategories, the small sample size in some categories may limit the statistical significance of the conclusions. For instance, the 'Move' subcategory contains only a few dozen problems, making performance metrics for it potentially less stable and prone to significant variance.

- While the multiple-choice question  format ensures evaluation objectivity, it also simplifies the task and cannot fully rule out the possibility of models succeeding through random guessing. Furthermore, unlike free-form answers, this format cannot provide deeper insights into the specific flaws within a model's reasoning process when it makes an error.

- The paper states that the data originates from "online elementary school mathematics question banks" (p. 4) but fails to specify their sources (e.g., country of origin, curriculum followed). Different educational systems may vary in their presentation styles, diagrammatic conventions, and logical focus for visual problems. Although the problems were translated into English, their inherent pedagogical bias might persist, potentially affecting model performance on certain types of visual representations.

- While creating a single collage from multiple images (Fig. 3, p. 4) is an ingenious comparative experiment, the stitching process itself introduces new, undiscussed confounding variables: the layout, spacing, and relative size of the sub-images. This spatial arrangement could provide the model with additional visual or spatial cues not present in the original problem. Conversely, a cluttered collage could increase the model's parsing burden. Therefore, the performance improvement in the single-image setting may stem partially from these layout cues, rather than being solely a reflection of the model's ability to "integrate separate inputs."

**Questions:**

same as weakness

---

### Official Review · Reviewer_wgrm · 2025-10-30

**Soundness:** 2
**Presentation:** 2
**Contribution:** 2
**Rating:** 2
**Confidence:** 4

**Summary:**

The paper introduces VCBENCH, a benchmark for evaluating multimodal models through explicitly visual-dependent mathematical reasoning. It reformulates elementary-level math tasks into multimodal question–answer pairs across six cognitive domains, requiring integration of information from multiple images. The authors benchmark 26 MLLMs, highlighting their limited performance and difficulty in combining visual and symbolic reasoning.

**Strengths:**

1. The paper introduces a vision-centric benchmark that focuses on perceptual reasoning rather than knowledge-based problem solving. It emphasizes explicit visual dependency, encouraging models to reason about mathematical concepts through visual understanding.

2. The paper designs tasks that require multi-image integration, with each question containing an average of 3.9 images. This setup compels models to combine information from multiple visual inputs, offering a more realistic evaluation of multimodal reasoning beyond single-image perception.

3. The paper conducts a large-scale evaluation across 26 state-of-the-art LVLMs and 17 subtasks spanning six cognitive domains, offering a broad and systematic assessment of current multimodal reasoning capabilities.

**Weaknesses:**

1. The paper offers limited novelty compared to existing multimodal math benchmarks. Its main distinction from MV-MATH lies only in the explicitly defined Temporal dimension, while MV-MATH covers a broader range of problem types. Moreover, MathVista already incorporates temporal reasoning, making it difficult to identify any substantive advancement in visual reasoning contributed by this work.

2. The paper draws questionable experimental conclusions due to an outdated and incomplete model selection. Most evaluated LVLMs are older, and the benchmark omits recent high-performing open-source reasoning models such as MM-Eureka and other vision-centric architectures. Hence, the claim that “no model exceeds 50% accuracy” is not representative or conclusive under the current landscape.

3. The paper provides insufficient clarity about dataset collection and licensing. The sources of the problems and images are not well-documented, and the paper does not specify whether proper licenses or permissions were obtained.

4. The paper lacks comparative evaluation with recently benchmarks, such as MathVerse, We-Math, and DynaMath.

**Questions:**

Clarify the dataset collection and licensing process, include comparisons with recent benchmarks such as MathVerse, We-Math, and DynaMath, and justify the exclusion of reasoning models like MM-Eureka to support the claim that current LVLMs all perform below 50%.

---

### Meta-Review · Area_Chair_ghhL · 2025-12-23

**Summary:**

Across reviews, the main value of the submission is a reasonably sized benchmark (VCBench) targeting elementary-level multimodal math problems with **explicit multi-image visual dependency**, accompanied by broad baseline results over 26 LVLMs. However, reviewers raised several concerns that collectively outweigh the contribution for ICLR:

(i) **limited novelty** relative to closely related multimodal math benchmarks (e.g., MV-MATH, MathVista/MathVerse and other recent suites), with unclear differentiation beyond minor task framing;

(ii) **incomplete and potentially outdated evaluation coverage**, missing several recent strong vision-reasoning / long-CoT / RL-enhanced models, which weakens the strength of the headline conclusion that “no model exceeds 50%”; and

(iii) **insufficient documentation of dataset acquisition and licensing**, as well as limited methodological detail about sourcing, annotation protocol, and bias/coverage.

Additionally, reviewers noted the analysis is often **descriptive rather than diagnostic**, with limited failure-case depth and limited actionable insight beyond reporting low accuracy.

**Reviewer Concerns:**

No author response was provided here, so none of the substantive concerns were addressed in discussion.

**Reviewer Scores:**

Given the lack of an author response in the record, I do not expect material score changes.

---

### Decision · Program_Chairs · 2026-01-26

Reject